# Promoting abnormal grain growth in Fe-based shape memory alloys through compositional adjustments

M. Vollmer[1], T. Arold[1], M.J. Kriegel [2], V. Klemm [2], S. Degener[1], J. Freudenberger[2,3] & T. Niendorf [1]

Iron-based shape memory alloys are promising candidates for large-scale structural applications due to their cost efficiency and the possibility of using conventional processing routes from the steel industry. However, recently developed alloy systems like Fe–Mn–Al–Ni suffer from low recoverability if the grains do not completely cover the sample cross-section. To overcome this issue, here we show that small amounts of titanium added to Fe–Mn–Al–Ni significantly enhance abnormal grain growth due to a considerable refinement of the subgrain sizes, whereas small amounts of chromium lead to a strong inhibition of abnormal grain growth. By tailoring and promoting abnormal grain growth it is possible to obtain very large single crystalline bars. We expect that the findings of the present study regarding the elementary mechanisms of abnormal grain growth and the role of chemical composition can be applied to tailor other alloy systems with similar microstructural features.

[1] Institute of Materials Engineering, Universität Kassel, Mönchebergstraße 3, 34125 Kassel, Germany. [2] Technische Universität Bergakademie Freiberg, Institute of Materials Science, Gustav-Zeuner-Straße 5, 09599 Freiberg, Germany. [3] IFW Dresden, Institute for Metallic Materials, Helmholtzstrasse 20, 01069 Dresden, Germany. Correspondence and requests for materials should be addressed to M.V. (email: vollmer@uni-kassel.de)

The control of grain size is essential for many materials and their industrial applications. For example, the absence of grain boundaries is sought in many materials in order to reduce creep rates in nickel-based superalloys[1] or to produce wafers for the semiconductor industry[2]. In several shape memory alloy (SMA) systems, which can exhibit large recoverable strains due to a thermoelastic transformation, many attempts have been made in order to achieve superior mechanical properties, such as better formability, a lower sensitivity against intergranular cracking, and a higher fatigue strength, by reducing grain size[3–9]. In the past decade ultra-fine grained and nanocrystalline structures were found to strongly promote the functional stability of Ni–Ti[10–16]. This was attributed to hardening of the matrix, changes of lattice parameters, and different selection of the martensite variants[12,14–16]. However, other SMA systems show significant improvements of the functional properties only if the grains completely cover the entire cross-section of the samples, for example, copper-based SMAs[17–23], cobalt-based SMAs[24,25], and iron-based SMAs[26–35]. Investigations by Ueland and Schuh[17–19] showed that grain boundary triple junctions as well as large grain boundary fractions have a detrimental effect on the pseudoelastic performance of polycrystalline Cu–Zn–Al and Cu–Al–Ni. They found that anisotropic transformation strains lead to large incompatibilities at the grain boundaries and to stress concentrations in the vicinity of triple junctions, resulting in multi-variant martensite morphologies, which are detrimental to a good shape memory performance. In contrast, oligocrystalline structures, also referred to as bamboo structures, showed superior pseudoelastic properties, since grain boundary areas were minimized and grain constraints caused by triple junctions were avoided. However, it is difficult to achieve such coarse grain structures by normal grain growth. Furthermore, conventional techniques to grow single crystalline materials like the Bridgman–Stockbarger[36,37] or the Czochralski[38] technique are complex, cost intensive, and bear limitations with respect to dimensions as well as shape.

Recently, Omori et al.[39] introduced a promising process to control the grain size by a new kind of abnormal grain growth (AGG) in a Cu–Al–Mn SMA. AGG, sometimes also referred to as secondary recrystallization[40,41], is characterized by a rapid growth of some particular grains, growing larger than surrounding grains and eventually leading to a bimodal or even to an oligocrystalline or single-crystalline grain structure[1,42–47]. In general, the circumstances under which AGG occurs are manifold[1,48]; however, the phenomenon has usually been investigated under static annealing conditions[49–54]. In the past decade AGG was observed during high-temperature plastic deformation of molybdenum and tantalum[46,48,55–57]. It was shown that AGG under these dynamic conditions leads to very large abnormal grains[46,48,58]. In consideration of this well-established differentiation of AGG mechanisms, that is, static and dynamic AGG, the new kind of AGG observed by Omori et al.[39] was assigned to dynamic AGG due to its underlying mechanisms[58]. Coarse grain morphologies in the Cu–Al–Mn SMA were obtained by a cyclic heat treatment (CHT) between a single-phase and a two-phase region, which was related to the formation of subgrain structures. It was assumed that the loss of coherency during growth of a semi-coherent second phase leads to the formation of subgrains in the parent phase[39,59,60]. The newly formed subgrains remain in the parent phase even after dissolution of the second phase and provide the necessary energy for particular grains to grow abnormally. Thereby, the surrounding grains are assimilated by the abnormally growing grains and no subgrain structures are observable in the grown areas anymore[39,60]. Recently, similar observations were made in a Cu–Al–Mn–Mo SMA[61]. However, in contrast to the AGG induced by a CHT, the grain growth in the Cu–Al–Mn–Mo SMA was promoted by the dissolution of nanoprecipitates leading to a continuous misorientation gradient within the grains.

Until now, AGG induced by a CHT was observed for Cu–Al–Mn[39,60] and for Fe–Cr–Co–Mo[62,63], as well as for Fe–Mn–Al–Ni[33,59,64–66]. It is thought that it will be feasible to adapt the CHT to further alloy systems showing a semi-coherent second phase. However, up to now, the underlying mechanisms are not fully understood and the high number of influencing factors for the CHT opens up many possibilities for further research. Recently, Kusama et al.[60] were able to obtain large single crystalline Cu–Al–Mn bars with a length of 700 mm and a diameter of 15 mm by varying the holding temperatures in the single-phase region between the cycles. Thereby, it was possible to increase the misorientation of the subgrain boundaries leading to higher sub-boundary energies and therefore to higher driving forces for AGG, finally resulting in large single crystal structures.

One of the alloy systems of particular interest for AGG is Fe–Mn–Al–Ni–X (X = Ti, Cr)[33,66,67], since it is a cost-efficient alternative to Ni–Ti and copper-based SMAs and a promising candidate for the realization of new material-intensive applications like damping elements for bridges and skyscrapers based on SMAs. Some of the advantages of this alloy system are the availability of relatively simple processing routes originating from steel industry and the low material costs, as well as the low slope of the Clausius–Clapeyron relationship (0.53 MPa K$^{-1}$)[33] over a wide temperature range. The thermoelastic martensitic transformation originates from the fine precipitation of an ordered, coherent phase in a disordered parent phase[33,68]. Such nanoprecipitates are known to strengthen the matrix via formation of coherent stress fields and to reduce the temperature hysteresis by several hundred Kelvin[69–72]. In a considerable number of studies the influence of these fine β precipitates (B2, bcc) in Fe–Mn–Al–Ni on the martensitic transformation, the shift of transformation temperatures, and the functional properties was investigated[65,68,73–78]. However, only a few studies focused on the AGG behavior of this system[35,59]. Considering the experimentally determined grain boundary migration rates for normal heat treatment cycles in Fe–Mn–Al–Ni (=2.5 × 10$^{-6}$ m s$^{-1}$)[59] and in Cu–Al–Mn (=1.6 × 10$^{-5}$ m s$^{-1}$)[60], it is obvious that the grain boundary migration rates of these alloy systems differ by almost one order of magnitude. Therefore, coarse single crystals of similar size as for the Cu–Al–Mn SMA can hardly be obtained in the more cost-efficient Fe–Mn–Al–Ni alloy system.

In this study, we investigate the influence of the chemical composition on the AGG behavior of the Fe–Mn–Al–Ni–X system. We show that the addition of small amounts of titanium (1.5 at%) is suitable to increase the AGG migration rate drastically. Thereby, grain boundary migration rates similar to Cu–Al–Mn, being the highest reported so far, are achievable. Moreover, it is possible to obtain large single crystalline bars with a diameter of 6.3 mm revealing good pseudoelastic properties up to 8% applied strain. Considering the decreased quenching sensitivity of Fe–Mn–Al–Ni–Ti[66], an industrial application of the Fe–Mn–Al–Ni–X system seems to be highly promising. Furthermore, the grain boundary migration rate can also be strongly inhibited by the addition of small amounts of chromium (3.0 at%) demonstrating a wide adjustability of the grain boundary migration rate through the addition of small amounts of different chemical elements.

## Results

**Grain growth behavior of Fe–Mn–Al–Ni–X (X = Ti, Cr).** The recoverability of Fe–Mn–Al–Ni strongly depends on the grain size in relation to the cross-section of the samples[33–35]. It is obvious from the stress–strain curves in Fig. 1 that the

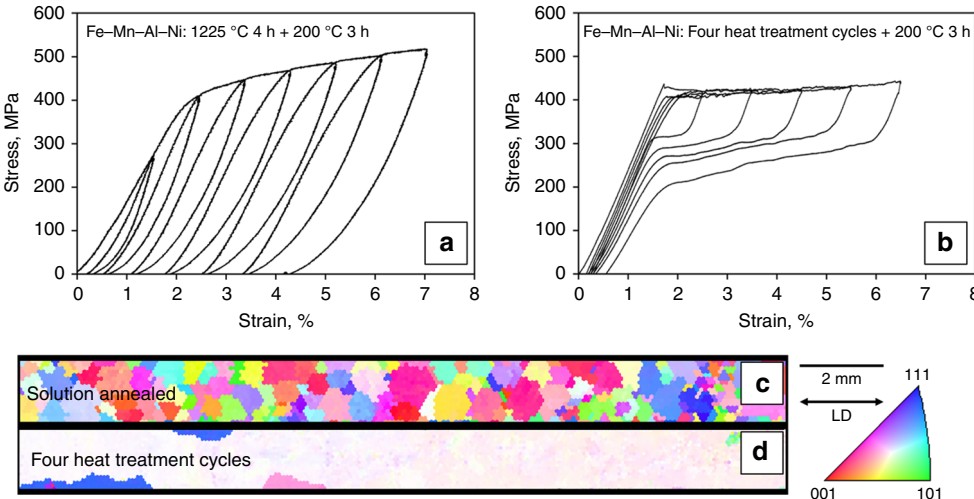

**Fig. 1** Pseudoelastic response of Fe–Mn–Al–Ni tension samples with different grain sizes. Stress–strain curves of Fe–Mn–Al–Ni after a solution treatment at 1225 °C for 4 h in **a** and after a cyclic heat treatment for four times in **b**. **c**, **d** show the corresponding electron-backscatter diffraction orientation maps of the samples plotted for the loading direction (LD). The colors correspond to the crystal directions given in the stereographic triangle. Details on the cyclic heat treatment procedure are shown in Supplementary Fig. 1a

solution-treated, polycrystalline condition exhibits poor pseudoelasticity, whereas the nearly single crystalline structure obtained by a CHT shows an almost perfect recoverability up to 7% strain. In order to investigate the influence of small amounts of titanium and chromium on the AGG behavior of Fe–Mn–Al–Ni–X, primarily the first cycle of a heat treatment between the α single-phase region at 1225 °C and the α + γ two-phase region at 900 °C, hereinafter referred to as single CHT, is considered in detail (cf. Supplementary Fig. 1a). The average grain diameter ($d_{3D}$), the maximum grain size, and the characteristic microstructure of the solution-treated samples (1225 °C 1 h) and samples undergoing one single CHT (1225 °C–900 °C–1225 °C) were analyzed in Fig. 2. The optical microscopy (OM) images show characteristic microstructures of (a) Fe–Mn–Al–Ni, (b) Fe–Mn–Al–Ni–Ti, and (c) Fe–Mn–Al–Ni–Cr after one single CHT. AGG occurred in Fe–Mn–Al–Ni as well as in Fe–Mn–Al–Ni–Ti, whereas grain growth in Fe–Mn–Al–Ni–Cr was impeded. Moreover, it is obvious from the micrographs that Fe–Mn–Al–Ni–Ti already shows a perfect bamboo structure upon the single CHT, whereas Fe–Mn–Al–Ni still possesses some small grains and triple junctions known to be detrimental to the pseudoelastic behavior. A quantitative analysis of the average grain diameter of solution-treated samples and samples after a single CHT is shown in Fig. 2d. The mean grain diameter of Fe–Mn–Al–Ni increased from about 650 μm after solution treatment up to 2300 μm after a single CHT. The grain size of the largest grain was about 5700 μm (cf. Fig. 2e). These values are in good agreement with results obtained by Omori et al.[59] for Fe–Mn–Al–Ni sheet specimens with 1 mm thickness undergoing a single CHT between 900 °C and 1200 °C. Focusing on the mean grain diameter of Fe–Mn–Al–Ni–Ti before (480 μm) and after (7200 μm) a single CHT, it becomes obvious that the grain boundary migration rate is strongly increased by the addition of titanium. The largest grain was about 12,000 μm. In contrast to Fe–Mn–Al–Ni–Ti, the mean grain diameter of Fe–Mn–Al–Ni–Cr (660 μm) after solution treatment does not differ from the mean grain diameter after the single CHT and the maximum grain size increased only slightly from 1400 to 1600 μm. However, no grains exceeded the cross-section of the samples, indicating the strongly hampered grain growth in this condition. Detailed analyses of the grain size distributions of representative samples of each condition, that is, solution treated

and after a single CHT, for the three different chemical compositions are shown in Supplementary Fig. 2.

**Volume fraction and morphology of the γ phase.** Omori et al.[59] found that the grain size after CHTs in Fe–Mn–Al–Ni is related to the volume fraction of γ phase in the α + γ two-phase region: the higher the volume fraction of γ, the larger the grains after CHT. Therefore, the γ-phase volume fraction for different holding temperatures in the α + γ two-phase region was investigated. The detailed sequences of the heat treatments are shown in Supplementary Fig. 1b. The samples were solution treated at 1225 °C for 30 min, cooled down with a rate of 10 K min⁻¹ to different temperatures (400–1150 °C), held for 15 min, and finally quenched into cold water. As it is revealed by the temperature–volume fraction graphs in Fig. 3a, all different chemical compositions show similar characteristic curves, that is, the γ-phase volume fraction increases with decreasing holding temperatures up to a maximum of 75–90%. Dilatometer measurements shown in Fig. 3b revealed much lower γ solvus temperatures for Fe–Mn–Al–Ni–Cr (1070 °C) and Fe–Mn–Al–Ni–Ti (970 °C) in comparison to Fe–Mn–Al–Ni (1130 °C). This decrease of the solvus temperatures in Fe–Mn–Al–Ni–Cr and Fe–Mn–Al–Ni–Ti can be explained by the impact of chromium and titanium both being α stabilizers[66]. It is obvious that differences in γ solvus temperatures have an influence on the γ volume fraction at 900 °C, the temperature that was selected for the two-phase region in the single CHT. The γ volume fraction of Fe–Mn–Al–Ni at 900 °C is 84%, whereas the volume fractions of Fe–Mn–Al–Ni–Cr (69%) and Fe–Mn–Al–Ni–Ti (56%) are much lower. At the first glance the results suggest that the lower grain boundary migration rate of Fe–Mn–Al–Ni–Cr might be related to the lower γ-phase content at 900 °C. However, further investigations focusing on a single CHT between 1225 °C and 800 °C (78% volume fraction of γ) showed no significant change in grain sizes as compared to solution-treated samples and samples undergoing a single CHT between 1225 °C and 900 °C. In addition, Fe–Mn–Al–Ni–Ti shows much higher grain boundary migration rates in spite of the low γ-phase content.

OM images of the γ-phase morphology after water quenching from 900 °C are shown in Fig. 3c–e. It is evident that the morphology of the γ phase in Fe–Mn–Al–Ni and

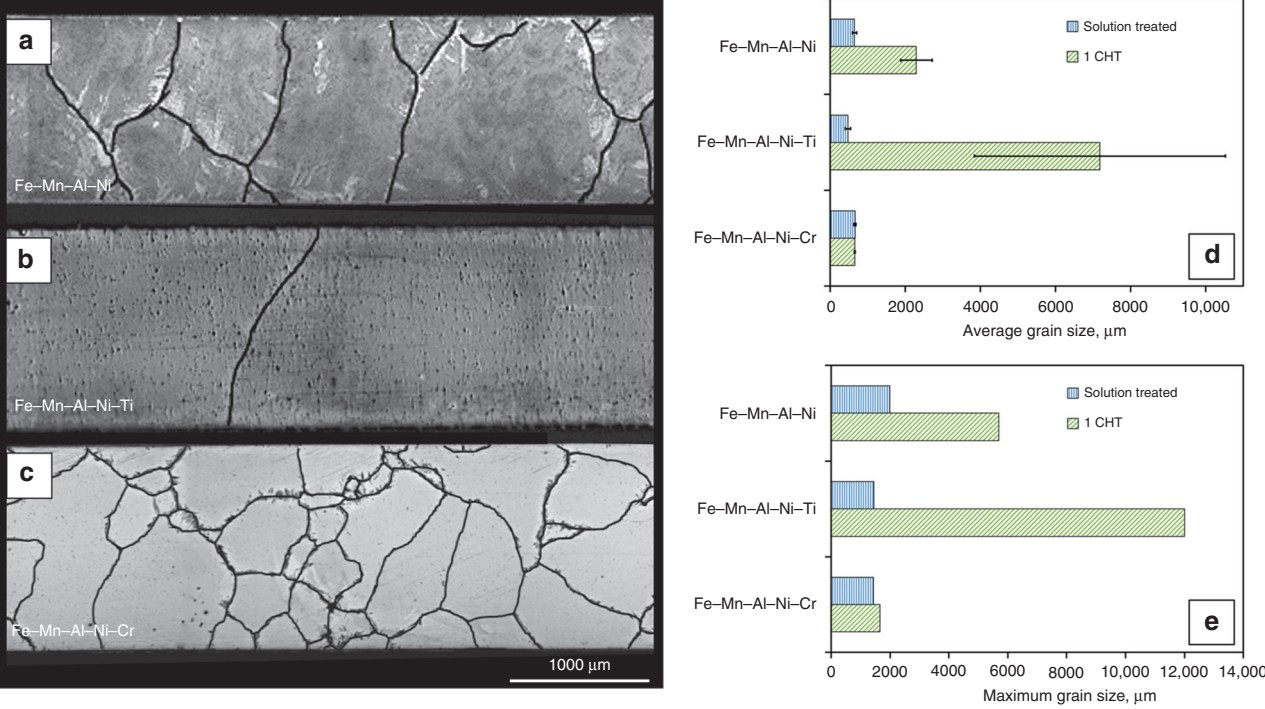

**Fig. 2** Evaluation of grain growth behavior of Fe–Mn–Al–Ni–X (X = Ti, Cr). Optical micrographs of the characteristic microstructure with highlighted grain boundaries after a single cyclic heat treatment (cf. Supplementary Fig. 1a) for (**a**) Fe–Mn–Al–Ni, (**b**) Fe–Mn–Al–Ni–Ti, and (**c**) Fe–Mn–Al–Ni–Cr. Average grain size (**d**) and maximum grain size (**e**) after solution treatment at 1225 °C for 1 h and after a single cyclic heat treatment(cf. Supplementary Fig. 1a). Error bars in **d** indicate the standard deviation for the investigated specimens

Fe–Mn–Al–Ni–Cr is similar in terms of size and overall appearance, whereas Fe–Mn–Al–Ni–Ti is characterized by a very fine distribution and an acicular structure of the γ phase. Electron-backscatter diffraction (EBSD) measurements were carried out in order to shed light on the role of the γ-phase morphology and fraction on the formation of the subgrain structures (cf. Fig. 4). The grain reference orientation deviation (GROD) maps show the characteristic misorientations within the α phase, clearly indicating the formation of subgrains[39,59,60]. It can be seen that the distances between the γ-phase lamellae are generally smaller and the overall microstructure appearance is more fragmented in Fe–Mn–Al–Ni–Ti, due to the γ-phase morphology shown before. However, there are no significant differences with respect to absolute values and distribution of the misorientation between all conditions.

**Influence of subgrain sizes and misorientations**. In order to investigate the characteristics of the subgrains, samples were cyclic heat treated between 1225 °C and 900 °C and immediately water quenched after final heating at 1225 °C. The detailed heat treatment sequence is shown in Supplementary Fig. 1c. The abnormally grown grains highlighted by the red lines in Fig. 5 reveal that the grain growth process in Fe–Mn–Al–Ni–Ti already finished, whereas the growth process was interrupted in Fe–Mn–Al–Ni and Fe–Mn–Al–Ni–Cr. It is very likely that the higher grain boundary migration rate in Fe–Mn–Al–Ni–Ti is partially related to the reduced γ solvus temperature, since the dwell time in the single-phase region increases for a given maximum temperature. In order to estimate the grain boundary migration rate of Fe–Mn–Al–Ni–Ti, two more samples were treated using the same CHT procedure, but with the modification of immediate water quenching after heating up to 1000 °C and 1100 °C, respectively. In both samples, no abnormally grown

grains were found, thus AGG occurs between 1100 °C and 1225 °C. The time for heating from 1100 °C up to 1225 °C is calculated to be 750 s (12.5 min). The largest distance between residual subgrains and the grain boundary was measured to be 13,775 μm resulting in an experimental grain boundary migration rate of $1.84 \times 10^{-5}\,\mathrm{m\,s^{-1}}$. Thus, the experimentally found grain boundary migration rate of Fe–Mn–Al–Ni–Ti is slightly higher than the experimental grain boundary migration rate of Cu–Al–Mn ($=1.6 \times 10^{-5}\,\mathrm{m\,s^{-1}}$)[60] for a similar single CHT and almost one order of magnitude higher than for Fe–Mn–Al–Ni ($=2.5 \times 10^{-6}\,\mathrm{m\,s^{-1}}$)[59]. Moreover, it should be noted that the grain boundary migration rate could be even higher, since the starting temperature of the AGG process as well as the finishing temperature at 1225 °C were only estimated based on available data.

The subgrain boundaries of all conditions are highlighted by the blue dotted lines in the EBSD image quality (IQ) maps in Fig. 6. From these lines it can be seen that the subgrain size of Fe–Mn–Al–Ni–Ti is small ($r_{s\ (\mathrm{Fe–Mn–Al–Ni–Ti})} = 11.6\,\mu\mathrm{m}$) in comparison to Fe–Mn–Al–Ni ($r_{s\ (\mathrm{Fe–Mn–Al–Ni})} = 54.6\,\mu\mathrm{m}$) and Fe–Mn–Al–Ni–Cr ($r_{s\ (\mathrm{Fe–Mn–Al–Ni–Cr})} = 35.3\,\mu\mathrm{m}$). This fact is probably related to the changed size and morphology of the γ phase in Fe–Mn–Al–Ni–Ti. The mean average misorientations of the subgrains were measured to be 0.88° for Fe–Mn–Al–Ni, 0.86° for Fe–Mn–Al–Ni–Ti, and 0.92° for Fe–Mn–Al–Ni–Cr. Obviously, all values are close to the value of 1° mentioned by Omori et al.[59] for Fe–Mn–Al–Ni. In consequence, the change in chemistry and the concomitant change of γ phase with respect to volume fraction and morphology have no significant impact on the subgrain misorientation characteristics.

Based on a model for grain growth processes in cellular microstructures[49,79], Omori et al.[59] calculated the total driving force for AGG ($\Delta G_{\mathrm{total}}$) taking into account the subgrain boundary surface energy $\gamma_s$ and the surface energy for

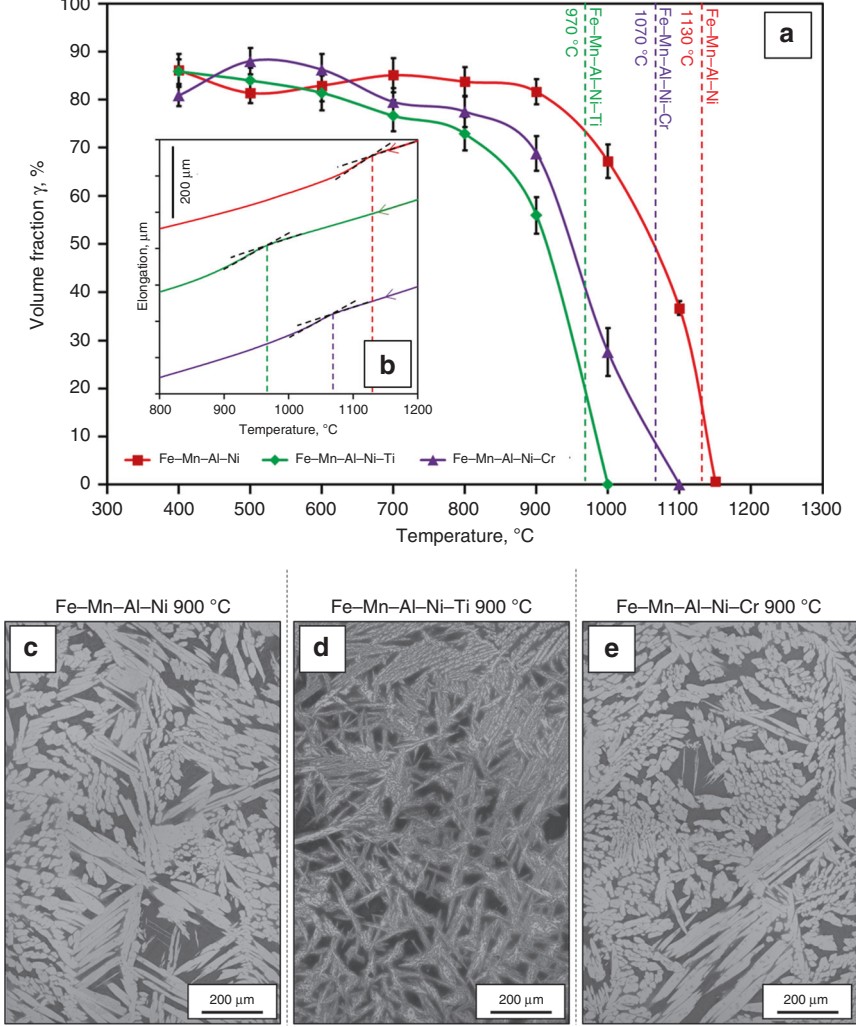

**Fig. 3** Volume fraction and microstructural appearance of γ phase in the two-phase region. **a** Volume fraction of γ phase after the heat treatment procedures shown in Supplementary Fig. 1b and corresponding γ solvus temperatures measured by dilatometer tests during cooling (**b**). Characteristic microstructures of Fe-Mn-Al-Ni (**c**), Fe-Mn-Al-Ni-Ti (**d**), and Fe-Mn-Al-Ni-Cr (**e**) after the same heat treatment procedure (cf. Supplementary Fig. 1b) with a lower temperature of 900 °C. Error bars in **a** indicate the confidence intervals for the investigated conditions

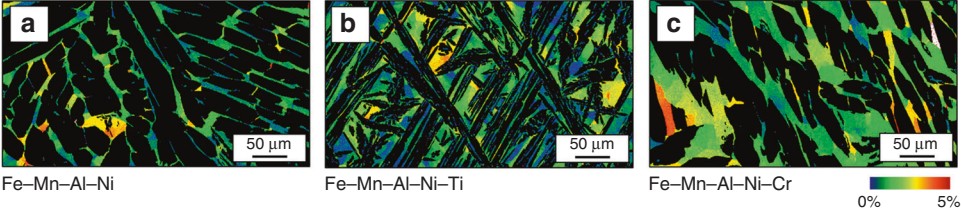

**Fig. 4** Grain reference orientation deviation maps for the α phase in Fe-Mn-Al-Ni-X (X = Ti, Cr). (**a**) Fe-Mn-Al-Ni, (**b**) Fe-Mn-Al-Ni-Ti, and (**c**) Fe-Mn-Al-Ni-Cr. The misorientation angles with respect to the average orientation of the grain are highlighted by the color code given. Details on the heat treatment procedure are shown in Supplementary Fig. 1b (lower temperature: 900 °C)

high-angle grain boundaries $\gamma_h$ (=0.617 J m$^{-2}$ based on Fe-3Si)[1]:

$$\Delta G_{total} = \Delta G_s + \Delta G_h = \frac{c_s \gamma_s V}{r_s} + \gamma_h V \left( \frac{c_n}{r_n} - \frac{c_a}{r_a} \right), \quad (1)$$

where $r_s$, $r_n$, and $r_a$ are the mean grain radii of the subgrains, the normal grains, and the abnormal grains, respectively. $V$ is the molar volume of the α phase (=7.366 × 10$^{-6}$ m$^3$ mol$^{-1}$)[33] and $c_s$, $c_n$, and $c_a$ are constants dependent on growth mode (two-dimensional (2D) or three-dimensional (3D)) and on the $r_n$–$r_a$ relation[1,79]. Assuming a homogeneous grain size after the first solution treatment at 1225 °C, that is, $r_n = r_a$, $c_n = c_a = 1$, and $c_s = 1.5$, it is obvious that the second part ($\Delta G_h$) of Eq. (1) becomes zero and that only the first part ($\Delta G_s$) contributes to the driving force that promotes AGG within the first heat treatment cycle. In further cycles, $r_a$ becomes much larger than $r_n$ ($r_a \rightarrow \infty$, $c_s = c_n = 1.5$, and $c_a = 1$) and the driving force for AGG increases.

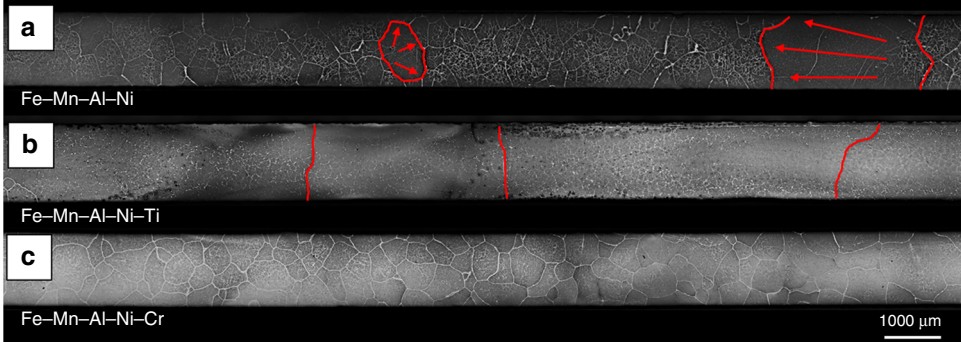

**Fig. 5** Grain growth behavior after one cyclic heat treatment. Optical micrographs of Fe-Mn-Al-Ni (**a**), Fe-Mn-Al-Ni-Ti (**b**), and Fe-Mn-Al-Ni-Cr (**c**) after the heat treatment detailed in Supplementary Fig. 1c. Grain boundaries of abnormal grown grains are highlighted by the red lines. Red arrows indicate the growth direction of the grains in Fe-Mn-Al-Ni

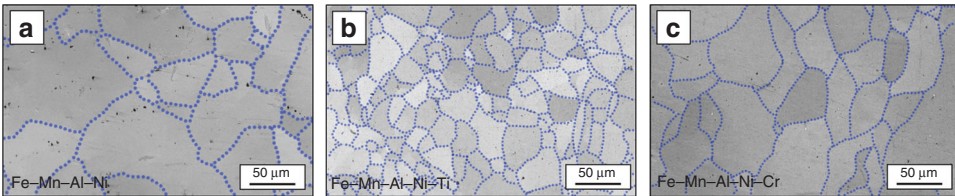

**Fig. 6** Subgrain structures characteristic for the Fe-Mn-Al-Ni-X (X = Ti, Cr) shape memory alloys. Electron-backscatter diffraction image quality maps of Fe-Mn-Al-Ni (**a**), Fe-Mn-Al-Ni-Ti (**b**), and Fe-Mn-Al-Ni-Cr (**c**). Blue dotted lines point out the low-angle grain boundaries of the subgrains. Details on the heat treatment are shown in Supplementary Figure 1c

In order to calculate the subgrain boundary surface energy ($\gamma_s$), the following Read–Shockley equation can be used:[59,60]

$$\gamma_s = \gamma_h \frac{\theta}{\theta_h} \left( 1 - \ln \frac{\theta}{\theta_h} \right), \tag{2}$$

where $\theta_h$ ($= 15°$)[59,60] is the critical angle defining a low-angle grain boundary and $\theta$ is the average misorientation of the subgrains. The relationship between the driving force $\Delta G_{total}$ and the grain boundary migration rate $dr$ per $dt$ is given by the following equation:

$$\frac{dr}{dt} = \frac{D^{gb}}{\delta RT} \Delta G_{total}, \tag{3}$$

where $D^{gb}$ ($= 5.33 \times 10^{-10}$ m$^2$ s$^{-1}$)[59] is the grain boundary diffusion coefficient, $\delta$ ($= 7.5 \times 10^{-10}$ m)[59] is the grain boundary thickness, $R$ ($= 8314$ J mol$^{-1}$ K$^{-1}$) is the gas constant, and $T$ is the temperature in Kelvin. All parameters for the investigated alloy systems and the results of the calculations are shown in Supplementary Table 1. The calculated grain boundary migration rates are $1.60 \times 10^{-6}$ m s$^{-1}$ for Fe–Mn–Al–Ni, $7.42 \times 10^{-6}$ m s$^{-1}$ for Fe–Mn–Al–Ni–Ti, and $2.56 \times 10^{-6}$ m s$^{-1}$ for Fe–Mn–Al–Ni–Cr, respectively. From the results obtained by the model, it is obvious that the calculated grain boundary migration rate of Fe–Mn–Al–Ni–Ti increased considerably in comparison to Fe–Mn–Al–Ni, which is due to the decreased subgrain size. This is in good agreement with the experimentally determined results and it can be concluded that smaller subgrain sizes, as a result of the altered chemical composition, are one of the key factors contributing to a higher driving force for AGG in Fe–Mn–Al–Ni–Ti.

Comparing the calculated migration rate of grain boundaries in Fe–Mn–Al–Ni–Cr ($2.56 \times 10^{-6}$ m s$^{-1}$) with the experimental results, it is obvious that the grain growth model does not correctly predict the value for Fe–Mn–Al–Ni–Cr such that other factors also seem to determine if AGG occurs or not. Figure 7 shows the characteristic microstructure of Fe–Mn–Al–Ni–Cr after a single CHT without annealing at 1225 °C (cf. Supplementary Fig. 1c). As highlighted by the dashed red lines, the subgrain

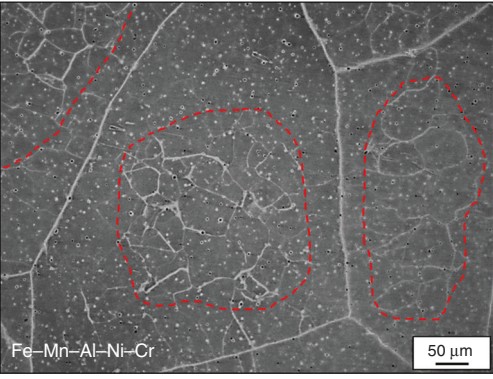

**Fig. 7** Characterization of the low density zones in Fe-Mn-Al-Ni-Cr. Optical micrograph of Fe-Mn-Al-Ni-Cr quenched after one single cyclic heat treatment (cf. Supplementary Fig. 1c). Large low-density zones are revealed between the high-angle grain boundaries and the subgrains in the grain interior highlighted by the red dashed lines

structures disappear in direct vicinity of the high-angle grain boundaries and are mainly restricted to the center of the grains. Kusama et al.[60] found similar low-density zones (LDZs) of subgrains in Cu–Al–Mn, although the LDZs were not as pronounced as in the Fe–Mn–Al–Ni–Cr. They assumed that the LDZs were responsible for the selection of the abnormally growing grains in Cu–Al–Mn, since the AGG process seems not to start before the LDZs have been overcome by normal grain growth of the high-angle grain boundaries. Presumably, the grains that first overcome the LDZs eventually have a growth advantage. Considering that AGG only starts if the high-angle grain boundaries reach the subgrains of the neighboring grains and taking into account that a relatively low grain boundary motion rate of the normal grains prevails, when the driving force for reducing the grain boundary area is no more sufficient[1], it is possible that AGG is strongly inhibited, if the LDZs cannot be

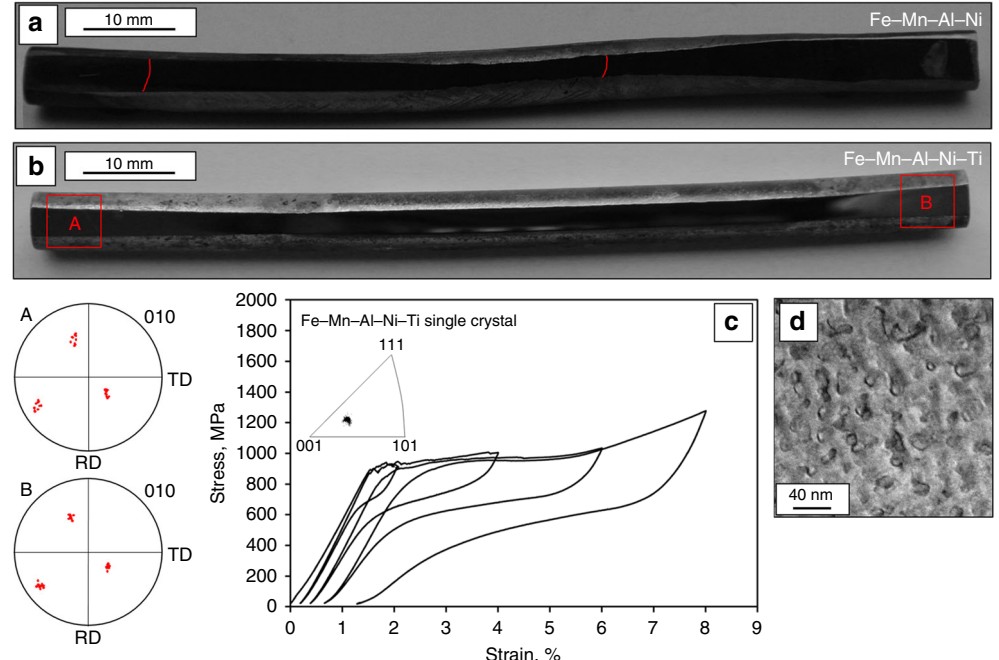

**Fig. 8** Abnormal grain growth in Fe–Mn–Al–Ni and a Fe–Mn–Al–Ni–Ti bars and corresponding pseudoelastic behavior of Fe–Mn–Al–Ni–Ti. Overview images of a Fe–Mn–Al–Ni bar (**a**) and Fe–Mn–Al–Ni–Ti bar (**b**), both featuring a length of 100 mm and a diameter of 6.3 mm, after the cyclic heat treatment procedure shown in Supplementary Figure 1d. Grain boundaries are highlighted by red lines. ⟨0 1 0⟩ pole figures were taken from the areas highlighted by the red squares on the Fe–Mn–Al–Ni–Ti bar. **c** Incremental strain test of a single crystalline Fe–Mn–Al–Ni–Ti compression sample up to 8% applied strain. The inset in **c** shows an inverse pole figure of the tested compression sample plotted with respect to the loading direction. **d** Characteristic scanning transmission electron microscopy image revealing the size of the β precipitates of the sample tested in **c**

overcome by normal grain growth. Fe–Mn–Al–Ni–Cr samples undergoing a single CHT (cf. Supplementary Fig. 1a) with an annealing time of 60 min at 1225 °C still show the same characteristic LDZs, that is, normal grain growth is strongly impeded such that the LDZs were not overcome by normal grain growth. In conclusion, it is likely that the strongly hampered AGG in Fe–Mn–Al–Ni–Cr is related to the relatively large LDZs in this condition. However, it may be possible to observe AGG in this condition, if the LDZs can be overcome by promotion of an appropriate heat treatment procedure.

**Abnormal grain growth in bars with 100 mm length.** In order to verify that the increased grain boundary migration rate of Fe–Mn–Al–Ni–Ti can be transferred to larger components, bars of Fe–Mn–Al–Ni and Fe–Mn–Al–Ni–Ti with a length of 100 mm and a diameter of 6.3 mm were subjected to the CHT process shown in Supplementary Fig. 1d. A combination of normal heat treatment cycles (between 900 °C and 1225 °C) and low-temperature heat treatment cycles was used. Kusama et al.[60] found that the driving force for AGG in Cu–Al–Mn was increased due to a higher misorientation of the subgrains as a result of repeated low-temperature heat treatment cycles. These findings were considered here and the upper temperatures of the low-temperature cycles employed were set only slightly above the solvus temperatures as identified by the dilatometer measurements, that is, 1150 °C for Fe–Mn–Al–Ni and 1000 °C for Fe–Mn–Al–Ni–Ti, respectively. Moreover, rates of the cooling and heating ramps were decreased to 1 K min⁻¹ in order to promote a more selective choice of abnormally growing grains[60]. Samples were investigated by means of OM after etching and EBSD in order to detect potential grain boundaries. Figure 8 shows the overview images of two representative bars after the cyclic heat treatment procedure with highlighted grain

boundaries. Three grains covering the whole cross-section were observed in the Fe–Mn–Al–Ni bar (a) indicating the relative low driving force for grain growth in this condition. In contrast, no grain boundaries were found in the Fe–Mn–Al–Ni–Ti bar (b). Furthermore, it can be deduced from the ⟨010⟩ pole figures that the grain orientation does not differ between both ends of the sample. Thus, a single crystalline structure with a length of 100 mm was obtained by AGG. The heat treatment procedure was repeated six times for both chemical compositions. In Fe–Mn–Al–Ni four samples consisted of three and more grains, one sample consisted of two grains, and one single crystalline bar was obtained. In Fe–Mn–Al–Ni–Ti one sample consisted of two grains, whereas five single crystalline bars were obtained. In order to demonstrate the pseudoelastic behavior of such single crystals, a compression sample was wire-cut by electro discharge machining from a single crystalline Fe–Mn–Al–Ni–Ti bar obtained by AGG. The orientation of the sample with respect to the loading direction is highlighted in the inverse pole figure shown as an inset in (c). It can be deduced from the incremental strain test (c) that the sample shows very good pseudoelasticity up to 8% strain. Good functional properties are also seen in constant amplitude cyclic tests; however, the role of remaining subgrain structures has to be carefully considered. Additional investigations shown in Supplementary Fig. 3 and Supplementary Fig. 4 and detailed in the Supplementary Discussion reveal the influence of subgrain structures on the pseudoelastic performance, as well as the functional fatigue behavior of Fe–Mn–Al–Ni–Ti. It has to be emphasized that no additional aging heat treatment was applied in between the solution treatment of the sample and the mechanical test. Recently, Poklonov et al.[80] showed fairly good pseudoelastic properties in solution treated, unaged ⟨122⟩ Fe–Mn–Al–Ni single crystals in tension as well as in compression. Moreover, Tseng et al.[73] found that precipitates are already existing immediately upon quenching of

Fe–Mn–Al–Ni single crystals into cold water. Therefore, it is likely that precipitates formed during the quenching process using 80 °C warm water. In order to evaluate the precipitate size, transmission electron microscopy (TEM) measurements were conducted and results are presented in Fig. 8d. It can be seen that β precipitates with sizes between 8 and 16 nm are present after quenching, which is in the same order of magnitude as precipitates being reported in Fe–Mn–Al–Ni[73]. Additional TEM investigations revealing the orientation relationship of the phases and chemical composition of β precipitates are shown in Supplementary Fig. 5.

## Discussion

In the current study, the influence of chromium and titanium on the AGG of Fe–Mn–Al–Ni–X was investigated. It was shown, that the addition of small amounts of a single element, that is, 1.5 at% titanium and 3.0 at% chromium, respectively, is suitable to either promote or inhibit AGG drastically. Based on the results, it was possible to reveal the prevailing mechanisms responsible for the enhanced grain boundary migration rate for AGG in Fe–Mn–Al–Ni–Ti and the decreased grain boundary migration rate in Fe–Mn–Al–Ni–Cr. In light of these findings, it seems to be feasible to transfer the idea of addition of small amounts of further elements to other alloy systems, showing AGG induced by a CHT, to tailor the AGG rates. Based on the analysis of different heat treatment conditions of Fe–Mn–Al–Ni, Fe–Mn–Al–Ni–Ti, and Fe–Mn–Al–Ni–Cr, it was revealed that the average grain size after one normal CHT between 900 °C and 1225 °C was three times higher in Fe–Mn–Al–Ni–Ti than in Fe–Mn–Al–Ni. The experimentally determined grain boundary migration rate of $1.84 \times 10^{-5}$ m s$^{-1}$ in Fe–Mn–Al–Ni–Ti is more than seven times higher than in Fe–Mn–Al–Ni. In contrast, the average grain size in Fe–Mn–Al–Ni–Cr hardly changed as compared to the solution treated condition. As α stabilizing elements, titanium and chromium shifted the γ solvus temperatures to lower values, that is, 970 °C for Fe–Mn–Al–Ni–Ti and 1070 °C for Fe–Mn–Al–Ni–Cr. As a result, a lower volume fraction of γ phase has been observed upon 900 °C annealing. However, the reduced volume fraction of γ phase seems not to have a strong influence on the AGG behavior in these conditions. Taking into account the morphology of γ phase, it was demonstrated that it was finer and more acicular in Fe–Mn–Al–Ni–Ti as compared to the morphologies seen in Fe–Mn–Al–Ni and Fe–Mn–Al–Ni–Cr. Thereby, a strongly decreased subgrain size was obtained resulting in a higher driving force for AGG. This is consistent with results calculated based on a model for grain growth processes in cellular microstructures. In contrast, it is very likely that the inhibited grain boundary migration rate in Fe–Mn–Al–Ni–Cr is imposed by formation of relative large LDZs of subgrains found in this condition.

Single crystalline bars of Fe–Mn–Al–Ni–Ti with a length of 100 mm and a diameter of 6.3 mm can be obtained robustly by AGG. Mechanical tests revealed good pseudoelastic properties up to 8% applied strain. No additional aging step was required. Single crystalline structures obtained by AGG as large as shown in the present work, that is, up to 220 mm in length (cf. Supplementary Fig. 7 and Supplementary Discussion), have never been reported for the Fe–Mn–Al–Ni–X alloy system before.

## Methods

**Specimen preparation.** Fe–34.0Mn–15.0Al–7.5Ni (at%), Fe–34.0Mn–15.0Al–7.5Ni–1.5Ti (at%), and Fe–34.0Mn–15.0Al–7.5Ni–3.0Cr (at%) ingots were produced by vacuum induction melting. Dog-bone-shaped tension samples with a gauge length of 18 mm and a cross-section of 1.6 mm × 1.5 mm, as well as bars with a length of 100 and 300 mm, respectively, all having a diameter of 6.3 mm, were used in this study. All samples were sealed into quartz tubes under argon atmosphere for the heat treatments. The sequences for the heat treatments

are shown in Supplementary Fig. 1. Fe–Mn–Al–Ni and Fe–Mn–Al–Ni–Cr samples for initial characterization (cf. Supplementary Fig. 1a), grain size investigations (cf. Supplementary Fig. 1a), and subgrain size investigations (cf. Supplementary Fig. 1c) were finally quenched into 80 °C warm water in order to prevent crack formation during quenching[64], whereas Fe–Mn–Al–Ni–Ti tension samples were finally air cooled since quenching sensitivity is significantly reduced by the titanium[66]. The volume fraction of the γ phase was investigated after heat treatments at different temperatures as shown in Supplementary Fig. 1b. For this purpose, all samples were quenched into 20 °C cold water in order to avoid changes in the γ-phase content during cooling. The Fe–Mn–Al–Ni and Fe–Mn–Al–Ni–Ti bars were subjected to the heat treatment shown in Supplementary Fig. 1d and subsequently quenched into 80 °C warm water. Compression samples with dimensions of 3 mm × 3 mm × 6 mm were wire-cut by EDM from a Fe–Mn–Al–Ni–Ti single crystal bar in order to investigate the pseudoelastic behavior. Before testing, the samples were again sealed into quartz tubes under argon atmosphere, solution annealed at 1225 °C for 30 min, and finally quenched into 80 °C warm water in order to avoid influences imposed by the EDM process. After the heat treatments, all samples were ground down to 5 μm grit size and vibro polished using colloidal SiO$_2$ suspension with 0.02 μm particle size for further investigations.

**Pseudoelastic tests.** The incremental strain tests using Fe–Mn–Al–Ni tensile samples have been performed using a miniature load frame equipped with a 10 kN load cell at a constant displacement rate of 5 μm s$^{-1}$. The mechanical tests using the Fe–Mn–Al–Ni–Ti compression samples were conducted using a servo hydraulic testing machine equipped with a 63 kN load cell at a constant displacement rate of 5 μm s$^{-1}$. Strains for the Fe–Mn–Al–Ni samples were calculated from displacement data and strains for the Fe–Mn–Al–Ni–Ti samples were measured using an extensometer with a 12 mm gauge length directly attached to the compression grips. For in situ measurements, a digital microscope equipped with a tele-zoom objective was mounted in front of the servo hydraulic testing machine.

**Microstructural observations.** EBSD measurements were conducted using a scanning electron microscope operated at 20 kV. For X-ray diffraction measurements, a diffraction angle 2θ of 124.9° using Mn-Kα radiation was used. The X-ray tube was operated at 35 kV and 30 mA and the measurement grid was 2° step size for tilt angle ψ and azimuth angle φ. The maximum tilt angle was 60°. A counting time of 0.5 s for each orientation was used. For OM, samples were etched using a solution of 50% HCl, 33% ethanol, 8.5% H$_2$O, and 8.5% CuSO$_4$. In order to investigate the β precipitates after solution annealing and quenching of the single crystalline Fe–Mn–Al–Ni–Ti sample, a TEM (operated at 200 kV) was utilized. For sample preparation, a thin lamella was extracted in $\langle 0\,0\,1 \rangle_a$ direction by the lift-out technique using a focused ion beam (FIB) system.

**Evaluation of grain sizes and grain size distribution.** Grain sizes and subgrain sizes were evaluated based on optical micrographs (grain sizes) and EBSD IQ maps (subgrain sizes), respectively, using the linear intercept method (ISO 643). The average length of the grains ($\bar{l}$) was evaluated by

$$\bar{l} = \frac{\sum L}{\sum N}, \tag{4}$$

where $\sum L$ is the cumulative length of the line segments drawn on the optical micrographs/EBSD IQ maps and $\sum N$ is the cumulative number of counted grains on all line segments for one condition. Several samples with eight line segments per sample were used for the grain size determination of each condition. The average length of the grains was evaluated based on $131 \leq \sum N \leq 2545$. EBSD IQ maps were used in order to evaluate the subgrain size. Six line segments were used for every EBSD IQ map and the number of cumulative grains was between 33 and 116. The relationship between the average length of the grains ($\bar{l}$) and the average grain radius ($r_{3D}$) was calculated according to Fullman[81] by assuming a spherical shape:

$$r_{3D} = \frac{1}{2}d_{3D} = \frac{3}{4}\bar{l}. \tag{5}$$

For investigations of grain size distributions, grain boundaries were determined from optical micrographs of characteristic samples. The area of each grain was evaluated using the software ImageJ. Afterwards, grains were sorted according to their area and grouped into area classes. The frequency was calculated from the number of grains in a class in relation to the number of all grains. The area fraction was calculated based on the ratio of the accumulated area of the grains in a class and the total area of the investigated sample.

**Calculation of the subgrains mean average misorientation.** The distributions of the GRODs were extracted from the EBSD measurements shown in Fig. 6 in order to calculate the mean average misorientations of the subgrains (θ) for the grain boundary migration rate model. The relative frequency of the misorientations ($h_{n \times 0.02°}$) was plotted over the misorientation angle ($\theta_{n \times 0.02°}$) with a step size of 0.02° up to 5° misorientation (cf. Supplementary Fig. 6). From this, mean average

misorientations were calculated as follows:

$$\theta = \sum_{n=0}^{250} \theta_{n \times 0.02°} \times h_{n \times 0.02°}. \qquad (6)$$

**Evaluation of γ-phase volume fraction.** The volume fraction of γ phase after the heat treatments, as shown in Supplementary Fig. 1b, was evaluated by the manual point counting method (ISO 9042). Forty square grids ($n$) with 16 points per grid ($P_T$) were superimposed to the micrographs of each sample in order to evaluate the arithmetic average of the volume fraction of γ phase ($\bar{P}_P$) in %:

$$\bar{P}_P = \frac{1}{n}\sum_{i=1}^{n}\frac{P_i}{P_T} \times 100. \qquad (7)$$

For each square grid, the number of points coinciding with the γ phase was counted ($P_i$). According to ISO 9042, points falling on the boundary as well as points with a doubt were counted as one-half. The 95% confidence interval (CI) was calculated as follows:

$$\text{CI} = \pm 2 \frac{s}{\sqrt{n}}, \qquad (8)$$

where $s$ is the standard deviation of the 40 square grids evaluated per sample.

**Dilatometer measurements.** Dilatometer measurements in argon atmosphere were conducted using samples with a diameter of 5 mm and a length of 50 mm. Samples were hold at 1225 °C for 15 min and subsequently cooled with a rate of 10 K min$^{-1}$ in order to measure the γ solvus temperatures.

## Data availability
The datasets generated and analyzed during the current study are available from the corresponding author on reasonable request.

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

## Acknowledgements

Financial support by German Research Foundation (project no. 400008732 (NI 1327/ 20–1)) is gratefully acknowledged. We would like to thank A. Liehr and L. Laabs for help with X-ray diffraction measurements and sample preparation.

## Author contributions

M.V. did pseudoelastic tests, EBSD investigations, cyclic heat treatments, and evaluation of grain sizes, as well as determination of the grain boundary migration rates. T.A. performed and evaluated heat treatments in order to characterize the volume fractions of γ phase and conducted dilatometer measurements. M.J.K. and V.K. prepared the FIB lamellae, conducted the TEM measurements, and evaluated the results. S.D. performed and evaluated the X-ray diffraction measurements. J.F. processed the initial Fe–Mn–Al–Ni as well as Fe–Mn–Al–Ni–Ti bars. T.N. and M.V. did design of experiments, interpreted data, and discussed the results. M.V. wrote the main manuscript. All authors revised and approved the manuscript.

## Additional information

**Competing interests:** The authors declare no competing interests.

