## [Peer Review File · Nature Communications]

Reviewers' comments:

Reviewer #1 (Remarks to the Author):

The abnormal grain growth (AGG) by cyclic heat treatment has first reported for Cu-Al-Mn shape memory alloy (SMA) in 2013 (Ref.23), already applied to the FeMnAlNi SMA in 2013 (Ref.23) and 2016 (Ref.24). The analysis on growth rate of the AGG, which is used in Discussion of this paper, was reported in 2018 and single-crystalline rods with 70cm length was achieved in Cu-Al-Mn SMA (Ref.25). On the other hand, the maximum size of single-crystal obtained for the FeMnAlNi SMA by the AGG is 30 mm length so far, because of a relatively small driving force for the AGG (Ref.24).

Under such background, this paper reports the drastic increase (100mm in length) of the maximum grain size in the FeMnAlNi SMA due to addition of 1.5%Ti and clarifies that the improvement essentially results from the decrease of the mean subgrain size, which is the dominant component of driving force for the AGG.

The paper is clearly and logically written and the result and discussion are well organized. The most important point of this article is the fact that the huge size of single-crystal over 10cm was first achieved by the new technique for ferrous alloy, which is one of the most important structural materials. As presented by authors, the driving force and the migration rate of AG are both larger than those in the Cu-Al-Mn alloy, which means that there is a possibility to obtain an ultra-large single crystalline rod with 70 cm length in the FeMnAlNiTi alloy as well as in the Cu-Al-Mn.

For acceptance, the authors are required to revise it for the following comments:

1. The addition of Ti brings about the precipitation of fine fcc phase, which results in the small subgrain structure. Why and how does Ti addition work to make the fcc precipitates fine? The solvus temperature of the fcc phase of Cr alloy is also lower than the conventional alloy, but the precipitate size is not so fine. I think the grain size of the fcc precipitate phase can be controlled by not only addition of the fifth element, but also some heat treatment, such as fast cooling and low temperature aging. Is the addition of Ti indispensable or not to obtain a huge single-crystal?
2. The electron diffraction pattern of Fig. S3(d) includes some satellite reflections, being different from that in the conventional alloy (Ref.32). Please give a comment on the origin of the satellite pattern.
3. I suppose the maximum single-crystalline length of 10cm is limited by the sample size used. If the author can show much larger length with a larger rod sample over 30 cm, the impact may further increase, because the size over 30 cm, at least, may be required for application to large components for buildings.
4. Were the samples presented in Fig.8 annealed in a quartz tube evacuated, as well as for the other small samples?

Ryosuke Kainuma

Reviewer #2 (Remarks to the Author):

The major claims of the present paper are:

- It introduces a method for tailoring the elementary mechanism of abnormal grain growth and the role of chemical composition, starting from the particular case of Ti or Cr additions at FeMnAlNi alloys, such as to render it applicable to other alloy systems with similar microstructural features.
- It argues that Ti additions refine the distances between γ -phase precipitates and reduce their orientation deviations.
- It has been assumed that grain growth rate increasing might be related to solvus temperature decrease.
- Decreasing subgrain size could be one of the key factors in driving force augmentation for

abnormal grain growth.

•The obtainment of the largest single crystalline structure, (ϕ 6.3 × 100 mm), by abnormal grain growth via cyclic heat treatment, opens up the way for further development of this new method.

The reported research effort is convincing, from the point of view of present reviewer, and after a minor revision effort, the results might be acceptable for publication.

If published, these results have the potential to largely influence the approach of further research by other people, in quest of a new chemical composition prone for abnormal grain growth, and associated superelastic behavior, after cyclic heat treatment.

Throughout the text you have used the terms "growth rate or "growth velocity", on one hand and "grain boundary motion rate" on the other hand. Moreover, you provided different values for these two parameters. Explain which is the difference between these two terms.

Please try to implement my suggested corrections and provide corresponding answers.

Reviewer #3 (Remarks to the Author):

This paper explores the concept of unconventional heat treatment to promote abnormal grain growth such that large grains with dimensions comparable to the sample width are obtained in Fe-based shape memory alloys. Usage of such so-called cyclic heat treatment between a single phase and a two-phase region has been proposed in a Science paper authored by others and has been applied to several shape memory alloy systems. This paper builds on prior work and performed more experiments on Fe-based shape memory alloys, but with the addition of Ti and Cr, and studied the role of these alloying elements on grain structure development. It is demonstrated that it is possible to obtain single crystalline Fe-based shape memory alloy specimen at a fast rate using this method with Ti as the alloying element. Therefore although this work is not scientifically novel, it is solid work and has a lot of engineering potentials. The paper is well written and the results are well presented.

The paper could be strengthened if

- (1) more work on the role of subgrain structure on pseudoelasticity is carried out;
- (2) multiple cycles of pseudoelastic curves at reasonably large strains are presented, as currently both Fig. 1(b) and Fig. 8(c) show incremental strain tests;
- (3) a thorough TEM study is performed to examine whether the cyclic heat treatment leads to nanoprecipitate development;
- (4) both quantitative and qualitative comparisons with literature work should be made. In the field of shape memory alloys many different attempts to engineer grains structures and engineer phases have been reported. In the field of grain growth and grain boundaries, a lot of work has been done on understanding abnormal grain growth;
- (5) a statistical analysis is performed for both specimens and grains as abnormal grain growth is not necessarily guaranteed.

NCOMMS-18-33815: Overcoming grain size limitations in Fe-base shape memory alloys through composition promoted abnormal grain growth

RESPONSE TO REVIEWERS COMMENTS

We thank all reviewers for their comments and useful suggestions. The changes we made are explained in detail in our response to the reviewers comments. The changes and additions made to address the reviewers' concerns are marked with **red font** throughout the manuscript.

Reviewers' comments:

Reviewer #1 (Remarks to the Author):

The abnormal grain growth (AGG) by cyclic heat treatment has first reported for Cu-Al-Mn shape memory alloy (SMA) in 2013 (Ref.23), already applied to the FeMnAlNi SMA in 2013 (Ref.23) and 2016 (Ref.24). The analysis on growth rate of the AGG, which is used in Discussion of this paper, was reported in 2018 and single-crystalline rods with 70cm length was achieved in Cu-Al-Mn SMA (Ref.25). On the other hand, the maximum size of single-crystal obtained for the FeMnAlNi SMA by the AGG is 30 mm length so far, because of a relatively small driving force for the AGG (Ref.24).

Under such background, this paper reports the drastic increase (100mm in length) of the maximum grain size in the FeMnAlNi SMA due to addition of 1.5%Ti and clarifies that the improvement essentially results from the decrease of the mean subgrain size, which is the dominant component of driving force for the AGG.

The paper is clearly and logically written and the result and discussion are well organized. The most important point of this article is the fact that the huge size of single-crystal over 10cm was first achieved by the new technique for ferrous alloy, which is one of the most important structural materials. As presented by authors, the driving force and the migration rate of AG are both larger than those in the Cu-Al-Mn alloy, which means that there is a possibility to obtain an ultra-large single crystalline rod with 70 cm length in the FeMnAlNiTi alloy as well as in the Cu-Al-Mn.

For acceptance, the authors are required to revise it for the following comments:

1. The addition of Ti brings about the precipitation of fine fcc phase, which results in the small subgrain structure. Why and how does Ti addition work to make the fcc precipitates fine? The solvus temperature of the fcc phase of Cr alloy is also lower than the conventional alloy, but the precipitate size is not so fine. I think the grain size of the fcc precipitate phase can be controlled by not only addition of the fifth element, but also some heat treatment, such as fast cooling and low temperature aging. Is the addition of Ti indispensable or not to obtain a huge single-crystal?

Answer: It is possible to obtain finer fcc precipitates by higher cooling rates or, as shown by Omori et al. (Omori, T. et al. Abnormal grain growth induced by cyclic heat treatment in Fe-Mn-Al-Ni superelastic

alloy. *Materials & Design* 101, 263–269 (2016)), through an aging treatment at 900°C after water quenching from 1200°C. However, up to now we were not able to obtain as large single crystals in Fe-Mn-Al-Ni as in Fe-Mn-Al-Ni-Ti. Tests conducted employed variation of heating and cooling conditions, e.g. experiments considering high cooling rates and low heating rates. However, not all experiments being published so far.

Observations similar to those in our experiments were made by Kusama et al. (Kusama, T. et al. Ultra-large single crystals by abnormal grain growth. *Nature Communications* 8, 354 (2017)) in Cu-Al-Mn. In their study it was shown that finer subgrain structures were obtained by higher cooling rates. However, at the same time the low density zones in the vicinity of the grain boundaries showed a smaller width. As a result, the grains started to grow abnormally at several points simultaneously eventually leading to a homogenous microstructure with relatively small grains.

At this point the role of Ti in the Fe-Mn-Al-Ni SMA has to be further evaluated in future work involving *i.a.* thermodynamic calculations, which are clearly out of the scope of the current work. From the microstructural observations it is assumed that a higher density of nucleation sites for fcc precipitation, eventually leading to the smaller subgrain size, is present due to titanium alloying. Furthermore, Ti addition is thought only to allow some particular grains to start to grow. This most probably is affected by the width of the low density zone. In consequence, the higher driving force as a result of the smaller subgrain size can be completely deployed to promote growth of a single grain. However, as mentioned before, details on the thermodynamic and kinetic aspects of nucleation and growth of the fcc precipitates clearly need to be addressed in a series of future studies. Work focusing on thermodynamics and time resolved characterization of phase transformation is in progress and will be published in follow-up studies.

2. The electron diffraction pattern of Fig. S3(d) includes some satellite reflections, being different from that in the conventional alloy (Ref.32). Please give a comment on the origin of the satellite pattern.

Answer: Due to variations in the chemical composition of the B2-type coherent particles (Al- and Ni-rich - Omori, T. et al. Abnormal grain growth induced by cyclic heat treatment. *Science* (New York, N.Y.) 341, 1500–1502 (2013); Tseng, L. W. et al. The effect of precipitates on the superelastic response of [100] oriented FeMnAlNi single crystals under compression. *Acta Materialia* 97, 234–244 (2015)) the crystal lattice becomes distorted and elastic stresses are induced in the material. The distortions between the B2-type particles and the parent phase lead to so-called Moiré contrasts in the TEM BF patterns. Furthermore, these regularly arranged contrasts lead to additional satellite reflections close to the diffraction spots in the SAED pattern. An explanation for the disappearance of the satellite reflections in Ref. 32 (Omori, T. et al. Microstructure and martensitic transformation in the Fe-Mn-Al-Ni shape memory alloy with B2-type coherent fine particles. *Appl. Phys. Lett.* 101, 231907 (2012)) might be a smaller sample thickness as compared to the thickness of the FIB lamellae (approx. 100 nm) of the present work. For the case that the sample thickness is close to the B2-type particle size, the lattices of B2 and the parent phase would not overlap and, thus, no Moiré contrast will be generated (disappearance of satellite reflections in Ref. 32).

(Supplementary files, Caption to Supplementary Figure 5)

3. I suppose the maximum single-crystalline length of 10cm is limited by the sample size used. If the author can show much larger length with a larger rod sample over 30 cm, the impact may further increase, because the size over 30 cm, at least, may be required for application to large components for buildings.

Answer: The authors already expected that even larger single crystals than 100 mm can be obtained in Fe-Mn-Al-Ni-Ti induced by AGG, however, only focused on samples of 100 mm length in the initial version of the paper for a meaningful comparison of the three different alloy compositions considered. In the light of the reviewer's comment, one bar with a length of 300 mm was cyclic heat treated in the same way the 100 mm bars were treated before. Thereby, two grains with a size of 220 mm and 80 mm, respectively, were obtained as revealed by means of optical microscopy and XRD texture measurements. At this point the authors would like to emphasize that it seems to be possible to further increase the maximum size of the single crystals by changing the heat treatment procedure, e.g. larger dwell times in the single phase region before quenching, lower temperatures in the two phase region, a higher number of cycles, different temperature ramps, or by applying thermo-mechanical treatments for texturing before AGG. However, this is out of scope of the present work and will be addressed in future studies.

The new results and the discussion were added to the Supplementary Files.

(Supplementary files, Page 3 and Supplementary Figure 7)

4. Were the samples presented in Fig.8 annealed in a quartz tube evacuated, as well as for the other small samples?

Answer: All samples were evacuated in quartz tubes for the heat treatments. In order to clarify this point, the authors decided to rephrase the following sentence in the Methods section: "Before testing, the samples were **again sealed into quartz tubes under argon atmosphere**, solution annealed at 1225°C for 30 min and finally quenched into 80°C warm water in order to avoid influences imposed by the EDM process."

(Manuscript, Page 14)

Reviewer #2 (Remarks to the Author):

The major claims of the present paper are:

- It introduces a method for tailoring the elementary mechanism of abnormal grain growth and the role of chemical composition, starting from the particular case of Ti or Cr additions at FeMnAlNi alloys, such as to render it applicable to other alloy systems with similar microstructural features.
- It argues that Ti additions refine the distances between γ -phase precipitates and reduce their orientation deviations.
- It has been assumed that grain growth rate increasing might be related to solvus temperature decrease.
- Decreasing subgrain size could be one of the key factors in driving force augmentation for abnormal grain growth.
- The obtainment of the largest single crystalline structure, (ϕ 6.3 × 100 mm), by abnormal grain growth via cyclic heat treatment, opens up the way for further development of this new method.

The reported research effort is convincing, from the point of view of present reviewer, and after a minor revision effort, the results might be acceptable for publication.

If published, these results have the potential to largely influence the approach of further research by other people, in quest of a new chemical composition prone for abnormal grain growth, and associated superelastic behavior, after cyclic heat treatment.

Throughout the text you have used the terms “growth rate or “growth velocity”, on one hand and “grain boundary motion rate” on the other hand. Moreover, you provided different values for these two parameters. Explain which is the difference between these two terms.

Answer: The terms “growth rate”, “growth velocity” and “grain boundary motion rate” were used as synonyms. The different values mentioned are stemming from differences between experimentally determined values and values calculated from the model. In order to avoid misunderstandings the authors decided to use the term “grain boundary migration rate” only. It was also double-checked whether all given values were marked as experimental or calculated values in the text and corrected when necessary.

(Manuscript, Changes throughout the text)

Please try to implement my suggested corrections and provide corresponding answers.
(Table attached as separate pdf-file).

Answer: The authors would like to thank the reviewer for his suggestions. For better readability, the text was revised as suggested by the reviewer. Changes are marked in red.

(Manuscript, Changes throughout the text)

Reviewer #3 (Remarks to the Author):

This paper explores the concept of unconventional heat treatment to promote abnormal grain growth such that large grains with dimensions comparable to the sample width are obtained in Fe-based shape memory alloys. Usage of such so-called cyclic heat treatment between a single phase and a two-phase region has been proposed in a Science paper authored by others and has been applied to several shape memory alloy systems. This paper builds on prior work and performed more experiments on Fe-based shape memory alloys, but with the addition of Ti and Cr, and studied the role of these alloying elements on grain structure development. It is demonstrated that it is possible to obtain single crystalline Fe-based shape memory alloy specimen at a fast rate using this method with Ti as the alloying element. Therefore although this work is not scientifically novel, it is solid work and has a lot of engineering potentials. The paper is well written and the results are well presented.

The paper could be strengthened if

(1) more work on the role of subgrain structure on pseudoelasticity is carried out;

Answer: The authors fully agree at this point. The role of subgrains on pseudoelasticity is an important topic, which needs to be carefully addressed. Single crystals obtained by AGG through a CHT consist of

areas with subgrain structures and areas without subgrain structures (Kusama, T. et al. Ultra-large single crystals by abnormal grain growth. Nature Communications 8, 354 (2017)). In order to provide first insights, a compression sample from an area with a high density of subgrain structures was wire-cut from the same Fe-Mn-Al-Ni-Ti bar that already had been chosen for extracting the compression sample shown in Figure 8c. This particular condition was chosen due to the expected high average misorientation of the subgrains (about 3°) as a result of the low-temperature cycles conducted before final solution annealing (cf. Supplementary Figure 1d - Kusama, T. et al. Ultra-large single crystals by abnormal grain growth. Nature Communications 8, 354 (2017)). Thus, we investigated a highly critical condition.

An *in situ* compression test up to 4 % strain was carried out and the results clearly reveal the impact of the highly misorientated subgrains on the general mechanical behavior as well as on the local microstructure evolution. Both, the results and the discussion of the results were added to the supplementary files and mentioned in the manuscript.

(Supplementary files, Page 1f. and Supplementary Figure 3; Manuscript Page 12).

In addition, a video of a bending test of a single crystalline Fe-Mn-Al-Ni-Ti bar was added to the supplementary files in order to reveal the overall performance of a large sample featuring areas with and without subgrain structures.

(Supplementary files, Video)

Further results reporting on the role of substructures on functional fatigue have been obtained as detailed in the answer to the following reviewer comment. Clearly, future studies need to focus on reducing the impact of subgrain structures, i.e. trying to keep the areas with subgrain structures as small as possible or to dissolve the subgrains in a further thermo-mechanical step.

(2) multiple cycles of pseudoelastic curves at reasonably large strains are presented, as currently both Fig. 1(b) and Fig. 8(c) show incremental strain tests;

Answer: In order to characterize the functional fatigue behavior of Fe-Mn-Al-Ni-Ti, two compression samples were cyclically loaded up to 4 % applied strain. One sample was extracted from an area with a high density of subgrain structures and the second one was extracted from an area without subgrain structures. It is obvious from the results shown (Supplementary files, Supplementary Figure 4) that functional fatigue already sets in in the very first cycle of the sample featuring subgrains, whereas the sample without subgrains shows a comparatively good functional fatigue behavior. In-depth analysis of the degradation behavior of Fe-Mn-Al-Ni-Ti and a comparison with the degradation behavior found for Fe-Mn-Al-Ni, being already studied by the current authors (Vollmer, M. et al. Cyclic degradation in bamboo-like Fe–Mn–Al–Ni shape memory alloys — The role of grain orientation. Scripta Materialia 114, 156–160 (2016), Vollmer, M. et al. Cyclic Degradation Behavior $\langle 001 \rangle$ -Oriented Fe–Mn–Al–Ni Single Crystals in Tension. Shap. Mem. Superelasticity 3, 335–346 (2017)) will be subject of future work.

The results and discussion on fatigue are included in the Supplementary Files and referenced in the manuscript.

(Supplementary files, Page 2f. and Supplementary Figure 4; Manuscript, Page 12)

(3) a thorough TEM study is performed to examine whether the cyclic heat treatment leads to nanoprecipitate development;

Answer: In order to examine whether the cyclic heat treatment leads to the development of nanoprecipitates in Fe-Mn-Al-Ni-Ti, additional TEM investigations were conducted on a sample, which was only solution annealed at 1225°C for 30 min and subsequently quenched in 80°C warm water, i.e. on a sample which was not cyclic heat treated. A FIB lamella was extracted in the same direction as done for the sample already characterized and included in the initial manuscript (cf. Supplementary Figure 5), i.e. $\langle 001 \rangle_{\alpha}$, by the lift-out technique using the same focused ion beam (FIB) system and investigations were conducted using the TEM system used before. As it is shown in the TEM micrographs below, the same nanoprecipitates, which were already shown in Supplementary Figure 5 after a CHT, were found. These observations are consistent with results found in Fe-Mn-Al-Ni single crystals produced by the Bridgman method, where the nanoprecipitates also occurred after solution annealing (Tseng, L. W. et al. The effect of precipitates on the superelastic response of [100] oriented FeMnAlNi single crystals under compression. *Acta Materialia* 97, 234–244 (2015)). Since the results clearly reveal that the nanoprecipitates also exist in samples not cyclic heat treated, it is not expected that the CHT has an influence on the nanoprecipitate development.

(4) both quantitative and qualitative comparisons with literature work should be made. In the field of shape memory alloys many different attempts to engineer grains structures and engineer phases have been reported. In the field of grain growth and grain boundaries, a lot of work has been done on understanding abnormal grain growth;

Answer: A lot of efforts have been made to achieve better properties in SMAs by tailoring grain structures and phases in SMAs. Most of the studies focused on small or even nanocrystalline grains and results of great importance and interest for the field were obtained. However, due to high anisotropy and a limited number of martensite variants able to accommodate the strain, some alloy systems require large grain sizes for good functional and mechanical properties. To further shed light on considerations

for distinct microstructure design in SMAs, a more detailed review of the SMA literature now is included in the introduction.

(Manuscript, Page 2)

In the field of abnormal grain growth a lot of important work has been made and various conditions and materials have been found being prone to AGG. In order to better distinguish the presented results from well-known AGG phenomena, the introduction was extended with regard to previous findings focusing on several aspects of abnormal grain growth.

(Manuscript, Page 2f.)

(5) a statistical analysis is performed for both specimens and grains as abnormal grain growth is not necessarily guaranteed.

Answer: Clearly, a statistical analysis of the effect of AGG on the grain size is highly important. In the initial version of the manuscript, a cumulative number of grains of several samples between 131 and 2545 already was considered when comparing the grain sizes after a solution annealing and a single CHT to ensure that the improved AGG response of Fe-Mn-Al-Ni-Ti and the impeded growth of Fe-Mn-Al-Ni-Cr is a significant finding. In order to investigate the grain size distribution within the samples in more detail and, thus, to address criticism, the authors decided to evaluate grain areas in samples of the solution treated conditions and of the single cyclic heat treated conditions, respectively, for each chemical composition. The resulting grain size distributions again confirm the more pronounced grain growth in Fe-Mn-Al-Ni-Ti and the impeded grain growth in Fe-Mn-Al-Ni-Cr. The results and discussion were added to the Supplementary Files and they are now referenced in the manuscript.

(Supplementary files, Page 1 and Supplementary Figure 2; Manuscript, Page 6)

Moreover, in order to check if the growth of single crystalline bars with a length of 100 mm is repeatable, six bars (for each case) of the two chemical compositions showing good AGG behavior, i.e. Fe-Mn-Al-Ni and Fe-Mn-Al-Ni-Ti, were cyclic heat treated. In Fe-Mn-Al-Ni-Ti five single crystals were obtained by AGG and one sample with two grains. On the contrary, four bars contained three and more grains, one bar contained 2 grains and one bar was found to be single crystalline in case of Fe-Mn-Al-Ni. These results are additionally provided in the revised manuscript.

(Manuscript, Page 11)

REVIEWERS' COMMENTS:

Reviewer #1 (Remarks to the Author):

The authors have well responded to my questions and criticism. Although no single crystal rod with 300mm in length can be obtained, 220mm has an enough impact to practical applications and shows a possibility to further improvement. The effect of subgrains on the superelastic behavior, which has been added in the revised version, is very important, making this article more valuable.

Reviewer #2 (Remarks to the Author):

Considering the purpose of the paper, the intense effort spent for the revision, the answers given to my comments and implementation of most of my suggested corrections, I think the article can be published under its present form.

NCOMMS-18-33815A: Overcoming grain size limitations in Fe-base shape memory alloys through composition promoted abnormal grain growth

RESPONSE TO REVIEWERS COMMENTS

We thank all reviewers for their commitment and their useful suggestions to improve the current study.

Reviewers' comments:

Reviewer #1 (Remarks to the Author):

The authors have well responded to my questions and criticism. Although no single crystal rod with 300mm in length can be obtained, 220mm has an enough impact to practical applications and shows a possibility to further improvement. The effect of subgrains on the superelastic behavior, which has been added in the revised version, is very important, making this article more valuable.

Reviewer #2 (Remarks to the Author):

Considering the purpose of the paper, the intense effort spent for the revision, the answers given to my comments and implementation of most of my suggested corrections, I think the article can be published under its present form.